# Structured multimaterial filaments for 3D printing of optoelectronics

Gabriel Loke[1,2,3], Rodger Yuan [1,2,3], Michael Rein [1,2,3], Tural Khudiyev[3], Yash Jain[4], John Joannopoulos[2,3,5] & Yoel Fink[1,2,3]

Simultaneous 3D printing of disparate materials; metals, polymers and semiconductors with device quality interfaces and at high resolution remains challenging. Moreover, the precise placement of discrete and continuous domains to enable both device performance and electrical connectivity poses barriers to current high-speed 3D-printing approaches. Here, we report filaments with disparate materials arranged in elaborate microstructures, combined with an external adhesion promoter, to enable a wide range of topological outcomes and device-quality interfaces in 3D printed media. Filaments, structured towards light-detection, are printed into fully-connected 3D serpentine and spherical sensors capable of spatially resolving light at micron resolution across its entire centimeter-scale surface. 0-dimensional metallic microspheres generate light-emitting filaments that are printed into hierarchical 3D objects dotted with electroluminescent pixels at high device resolution of $55\,\mu m$ not restricted by surface tension effects. Structured multimaterial filaments provides a path towards custom three-dimensional functional devices not realizable by existing approaches.

[1] Department of Materials Science and Engineering, Massachusetts Institute of Technology, Cambridge, MA 02139, USA. [2] Institute for Soldier Nanotechnologies, Massachusetts Institute of Technology, Cambridge, MA 02139, USA. [3] Research Laboratory of Electronics, Massachusetts Institute of Technology, Cambridge, MA 02139, USA. [4] Department of Electrical and Computer Engineering, Stony Brook University, Stony Brook, NY 11790, USA. [5] Department of Physics, Massachusetts Institute of Technology, Cambridge, MA 02139, USA. Correspondence and requests for materials should be addressed to Y.F. (email: yoel@mit.edu)

The design and fabrication of functional systems shaped in three-dimensional form factors can enable applications in diverse areas such as photonics[1], sensing[2,3], energy storage[4,5], and electronics[6,7]. This was recently achieved through approaches such as origami/kirigami[4,8], buckling mechanics[2,9], and self-assembly[7,10]. Comparatively, additive manufacturing, commonly known as 3D-printing, has seen rapid improvements in materials availability, unlocking its potential as an attractive, high-throughput technique towards forming 3D structures that are highly flexible in design and customized for different applications[11–14]. Having this ability to rapidly print highly-complex freeform shapes can potentially enable fully-tailored 3-dimensional electronic and optoelectronic devices for applications[15,16] in displays, wearable electronics, solid-state lightings, and biomedical devices. However, printing different material classes to create electronic devices is still a complex fabrication challenge in itself. This is because, principally, different print methods[13,17] were developed specifically for various material classes. For example, laser sintering is typically catered towards printing either metals or ceramics, while light-based print processes are commonly used for polymers. Other print processes, such as direct ink writing, fused filament fabrication, and ink-jet printing can deposit multiple materials in a single run from different nozzles via inks which individually contain metals[18], insulators[19,20], semiconductors[21]. However, even with the progress in ink materials and morphologies, printing multimaterial devices and sophisticated functional systems from these approaches remains limited. This limitation is attributed to multiple challenges including lack of interfacial bonding between different material classes[22,23] leading to delamination and poor device performance, limited device properties such as poor conductivity[17,24] arising from percolation threshold of viscoelastic inks, low device resolution caused by the sizeable nozzle diameter in facilitating extrusion[25] and wetting of inks due to surface tension during deposition[26], degradation of printed materials and distortion of structure geometry due to mismatch in melting temperatures of different materials during binder-removal post-annealing steps[27–29], and finally, the lengthy time[30] required for curing and annealing hence impeding scalability. These limitations in printing electronic devices have led to the development of alternative approaches to introduce system-scale functionalities, such as pick-and-place[31,32] sparse amounts of discrete print-incompatible devices into printed structures. Unfortunately, this approach is time-consuming since it necessitates interruption of the print process to embed each device component aggravated by the extensive time required for conductive ink annealing. In addition, this approach is restricted to larger device components in the millimeter-scale, limiting device resolution and achievable system geometries. Importantly, as the required device density, within these structures and object size, increases, these disadvantages become more pronounced.

In this work, we establish a fast, multiscale approach to print a diverse set of designable multimaterial filament-based inks to create complex 3-dimensional hierarchical functional systems that bridges micron-scale device resolution to centimeter-scale object size. In particular, unlike current composite inks[18–21] which have limited control over spatial localization of constituent materials, we demonstrate structured filaments that combine different interchangeable material classes with controlled interfaces while its internal materials can be microstructurally shaped into different topologies to enable varying ink functionalities.

## Results

**Filaments fabrication**. To fabricate these filaments, preforms with the desired multimaterial structure are thermally drawn into continuous kilometers-long microstructured filaments (Fig. 1a, b). The long-length scalability of the thermally-drawn filaments makes them ideal as feedstock for fused filament fabrication (FFF), or commonly known as fused deposition modelling (FDM), print technique[33]. Materials, independent of their melting points, can be jointly encapsulated (See "Methods" section for fabrication details) within a printable viscoelastic thermoplastic polymer cladding such as polycarbonate (PC) and cyclic olefin copolymer (COC), which serves as a print medium carrying multiple materials and an adhesion layer to build 3D macrostructures. Mediated by the encapsulating polymeric matrix, microscale materials are mechanically interfaced together to form high-quality device interfaces either during the thermal drawing of the filaments (between conducting polyethylene (CPE) and arsenic-selenide ($As_2Se_5$) in Fig. 1b), post-processing of the filaments (between bismuth-tin (BiSn) spheres with tungsten (W) wire and Zinc-sulphide (ZnS) in Fig. 1c) or during the print process (Fig. 1d). These electrically-linked interfaces, when printed, give rise to device functionalities in 3D objects (Fig. 1e, f).

**0D light-emitting filament ink**. We first validate transferring device functionality from the filament form to a 3D object through a printable pixelated electroluminescent light-emitting filament (Fig. 1a) which consists of a metallic BiSn core, an electrically-conducting W, an electroluminescent ZnS, and an insulating cladding PC surrounded by a print adhesion tie layer COC. Key to the formation of spatial light-emitters is the use of programmably-placed discrete microspheres (Fig. 1c). Hierarchically, the combination of the BiSn microsphere with ZnS and W enables 0-dimensional electroluminescent pixel within a larger 3D-printed macrostructure (Fig. 1e). To form these spheres, the BiSn core confined within the transparent viscoelastic PC matrix is subjected to laser-induced capillary breakup[34,35], which transforms the core into thermodynamically-stable BiSn spheres. As the sphere has a larger diameter than its core, the distance between the W and ZnS wires is bridged, forming interfacial electrical connections between BiSn and W, as well as, BiSn and ZnS (Supplementary Fig. 1, Fig. 2a, b). By connecting W and Cu to an alternating voltage source, these electrically conductive spheres link the electric potential from W towards the outer surface of ZnS, enabling sufficient electric field strength (Fig. 2c) to induce light emission from the ZnS layer via electroluminescence[36]. This demonstration of electrically-activated light emitters, arising from the microspheres in the filament, is the first stepping-stone to enable fabric and 3D-printed displays. We attain a maximum pixel density of 107 pixels-per-inch along the length of a 0.6 mm-thick filament (Supplementary Fig. 2). The width of light-emission reduces in size (Fig. 2d and Supplementary Fig. 2) with decreasing voltage magnitude and smaller sphere size (Supplementary Note 1). We define the pixel resolution by the size of the sphere, demonstrated to be as small as 55 μm which is notably on par with the pixel size of 55 μm in current super-retina high-definition displays.

**1D light-detecting filament ink**. With microspheres operating as 0D discrete functionality, we later illustrate a printable filament which contains a 1D-continuous functionality along the length of the filament. An axially symmetric photodetecting (PD) filament (Fig. 1b) comprised of semiconducting $As_2Se_5$, conducting CPE, and printable insulating PC, is thermally drawn. We made use of $As_2Se_5$ due to two reasons. First, $As_2Se_5$ is a semiconductor that has a bandgap which is sensitive to light across the visible range, enabling its use as an active material in visible-light photodetectors. Second, $As_2Se_5$ has matching glass transition and viscosity, that allows thermal co-drawability, with other low $T_g$

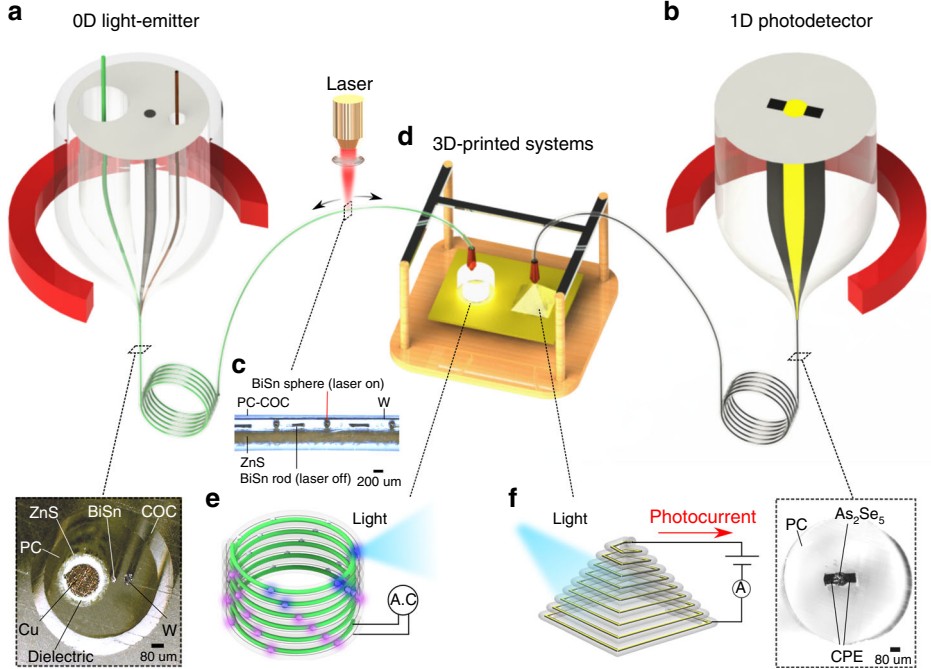

**Fig. 1** Designable structured multimaterial filament inks for three-dimensional printed functional systems. Thermal-drawing of multimaterial preforms into **a** 0-dimensional light-emitting and **b** 1-dimensional light-detecting filaments, with their corresponding cross-sectional optical micrographs of metal-insulator-semiconductor 3D microstructures. **c** Spatially-resolved laser-induced capillary formation of discrete BiSn spheres to form programmably-placed pixels within the light-emitting filament. **d** These microstructured filaments are fed into a regular fused filament fabrication printer with a modified nozzle, enabling the tailored formation of electrically-activated three-dimensional systems capable of spatial **e** light-emission and **f** light-detection from its entire structure

materials such as the insulating polycarbonate (PC) and electrically-conductive carbon-loaded polyethylene (CPE) to form a 1-dimensional flexible photodetecting filament device with a continuous detection capability along its length. The impingement of light onto the PD filament generates electric carriers in the $As_2Se_5$ core which are separated into the adjacent CPE electrodes under a potential difference (Fig. 2d). Having a confining PC matrix allows the interface between a polymer (CPE) and semiconductor ($As_2Se_5$) to be adjoined across long distances of the filament (as shown in Fig. 2f, g) thus achieving meters-length scale of 1D-continuous functionality. Upon light impingement, the photodetecting filament indicates the presence of a photo-current, which confirms photosensitivity of $As_2Se_5$ and electrical transport between CPE and $As_2Se_5$ (Fig. 2h).

**Filament surface heating**. Interfacing conductive and semi-conducting components before the print generates optoelectronic capabilities in the filament itself. Printing these microstructured filaments necessitates a reformulation of how the shape and functional interfaces of the intrinsic materials can be preserved during printing. In FFF, print temperatures typically range above the glass transition of the print material, so that materials are processed in a low viscosity state. In combination with mechanical stresses during print deposition, flow during wetting, and Rayleigh instability, intermixing and alterations to inner geometries generally occurs (Supplementary Figs. 3 and 4). To circumvent these challenges, we introduce a technique (Fig. 3a), termed *filament surface heating (FSH)* by which the structural integrity of interfaced microstructures is preserved. In this process, the multimaterial structured filament is fed through a hot end that provides heating localized to a short axial section (0.3 mm) of the filament. Higher print precision (up to 0.5 mm shown in this work) can easily be achieved by using

thermally-drawn filaments and hot ends of smaller diameters (Supplementary Note 2). Since the viscosity of polycarbonate below its glass transition temperature ($T_g$) is orders of magnitude higher than that above[37,38], keeping the polycarbonate layer surrounding the microstructures below its $T_g$ protects the functional microstructures from any print deformation. Yet, the filament surface must concurrently be viscous enough during the print for it to be deformable under stress. The temperature of the outermost thermoplastic adhesive promoter must be higher than its critical fusing temperature[39] ($T_{crit.}$) to create interdiffused linking polymeric chains between printed lines. To achieve such temperature conditions, we engineer a sharp radial temperature profile by setting a high hot end temperature with fast deposition speed. This effectively reduces the duration of heat diffusion into the filament, creating a skin effect in which the filament is only being surface-heated (Fig. 3b). We present the results of this approach in Fig. 3c, d which illustrates the assembly of micron features printed from the light-emitting and light-detecting filaments, respectively. In Fig. 3c, the cross-section depicts the fusing between the low $T_g$ (80 °C) viscous COC cladding, while the deposited multimaterial microstructure is able to maintain its full geometry from the original light-emitting filament due to shielding from the higher $T_g$ (147 °C) PC core. Similarly, through optimized print parameters (Supplementary Fig. 5), FSH enables the printed assembly of fused light-detecting lines which contain intimate interfaces between $As_2Se_5$ and CPE necessary for electrons and holes transfer (Fig. 3d). Finally, through the freeform capabilities of 3D-printing, printed objects can have tailored shapes, such as a star (Fig. 3e) printed from the light-emitting filament or a pyramid (Fig. 3f) printed from the light-detecting filament—both of which spanning dimensions into the centimeter scale while incorporating multimaterial microstructures at high resolution (~10 μm), which is not achievable by typical FFF (FDM) processes which have feature resolution of ~200 μm

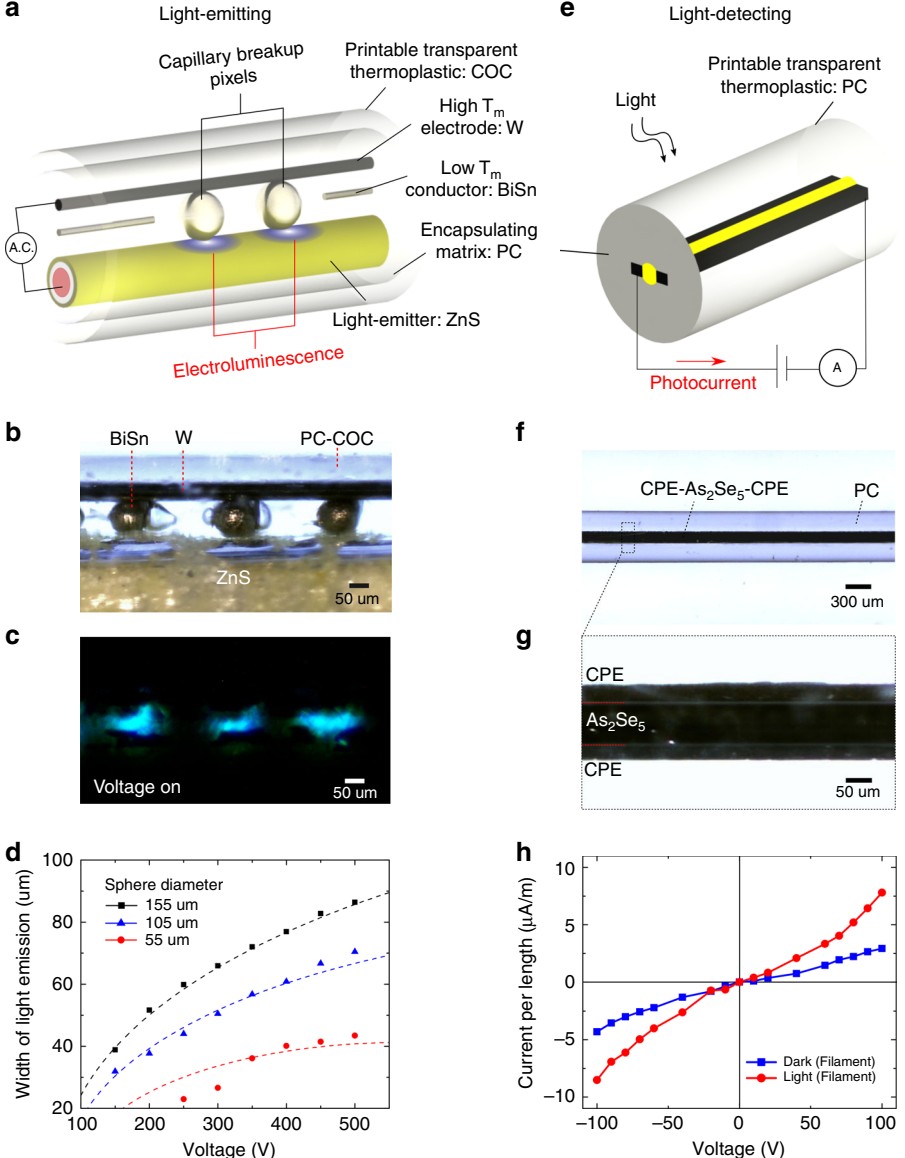

**Fig. 2** Characterization of the Optoelectronic Filaments. **a** Schematic of the pixelated light-emitting filament, illustrating the filament design for light-emission. **b** Optical micrograph of the in-filament low melting-point ($T_m$) BiSn conductive spheres electrically interfacing high melting-point W and ZnS. **c** Electroluminescent light-emission from ZnS at distinct sphere locations in (**b**). **d** Plot of width of light-emission versus AC voltage for different sphere sizes. The plotted dotted lines are derived from the theoretical framework presented in Supplementary Note 1. **e** Schematic of the light-detecting filament, with an external circuit connected to opposite ends of the different electrodes. **f**, **g** Planar optical micrographs of long-distance intimate electrical contact between CPE and $As_2Se_5$, which contributes to **h** generation of photocurrent upon light impingement. Both filaments are cladded by a printable thermoplastic (COC or PC) and contain functional interfaces between disparate materials classes immobilized by a PC matrix

limited by the nozzle dimensions and wetting. We later show in Fig. 3e, f, the ideal process parameters, namely the print speed and nozzle temperature, required to achieve both good polymeric adhesion and retainment of device functionality for the light-detecting and light-emitting filaments. Comparing the parameter map between both filaments, we note that the inclusion of a lower $T_g$ outer adhesion cladding around a higher $T_g$ core expands the speed-temperature area in which one achieves the desired strong adhesion with preservation of functionality. This strategy of elevating the glass transition temperature from the outer filament cladding to the inner core material reduces the failure of filament device printing, and can thus be implemented for future printing of multimaterial filaments with functionalities other than optoelectronics as described in the work.

**3-dimensional optoelectronic systems.** The opportunity to combine microscale materials within the filament offers the potential to incorporate abundant micro-devices in a printed macro structure. Leveraging our technique to spatially write lighted pixels, we showcase a printed cylinder (Fig. 4a) capable of displaying designed stripe patterns of light created by a collection of 90 microscale pixel-spheres (Fig. 4b, c), in which each lighted spot corresponds to 2 pixel-spheres. Unlike previous work[26] on printed light-emitting diodes, which has resolution in the order of mm due to wetting of low-viscosity inks from surface tension, the pixel resolution of our printed display is limited only by the sphere size and is shown to be in the microscale (55 μm). Furthermore, we demonstrate that the curved cylinder is capable of light emission around its geometry, offering a 360° continuous

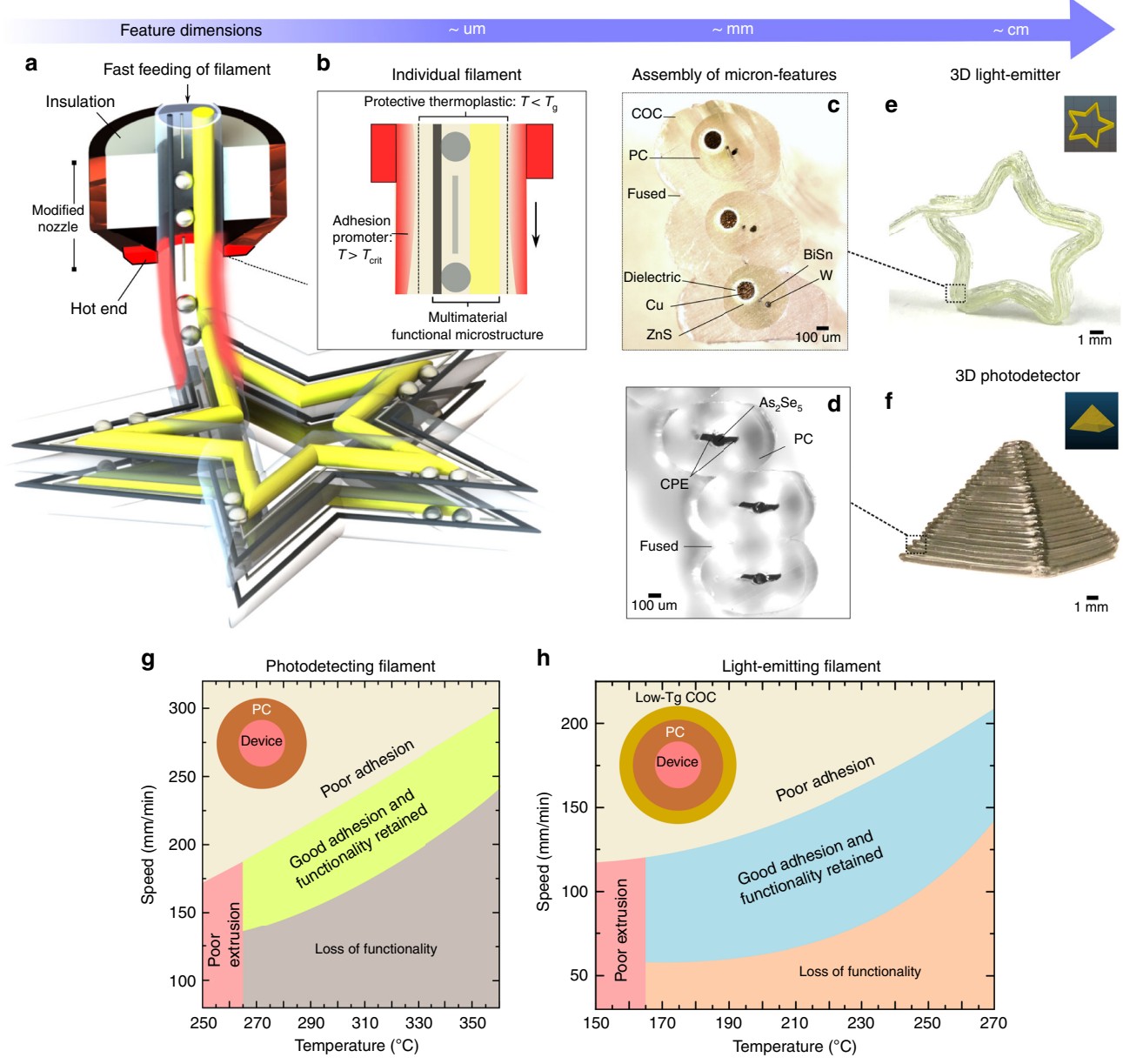

**Fig. 3** Multiscale Print Process. **a** The microstructured multimaterial filament is fed quickly through a short hot end. Precision of the print can be enhanced by using hot end and filaments of smaller diameters. **b** Filament surface heating (FSH) generates a surface-heated effect with the filament surface temperature rising beyond its critical fusing temperature ($T_{crit}$) while maintaining the temperature of the inner encapsulating polymer at a temperature below its glass transition ($T_g$). Microscale features within the ink can be precisely arranged for varying functionalities. **c**, **d** Cross-sectional optical micrographs showcasing a hierarchical assembly of millimeter-scale fused lines containing interfacial microstructures of different material classes. These assembly can be further stacked and shaped into three-dimensional centimeter-scale objects like **e** a star and a **f** pyramid which are printed from the light-emitting and light-detecting filaments, respectively, via FSH. Speed-temperature plots for the printing of **g** PC-cladded light-detecting and **h** COC-clad/PC-core light-emitting filaments

viewpoint which is largely applicable for 3D displays and robotics[40,41]. Moreover, in contrast to previous work[31,32] that requires a bulk platform for embedding devices and printing conductive ink interconnects, all interconnects and devices are already built into our filaments. Our one-step print approach reduces large amount of external connectorization and print time, enabling the quick formation of 3D objects (minutes) that can contain structurally-thin walls (order of mm) integrated with devices.

Inanimate objects seen around us are typically passive and only of aesthetic or structural use. This work expands the capabilities of such objects by endowing them with device functionalities. A patterned vase (Fig. 4d) printed with the photodetecting filament is demonstrated to be capable of detecting visible light, from its whole structure. It is noted that this vase is printed from a single continuous filament with the positive and negative voltage connections made across opposite ends of the printed filament and at different electrodes (Fig. 2e). The registering of a photocurrent (Fig. 4e) indicates that there is no contact between opposite CPE (short circuit) (Supplementary Fig. 4) or discontinuity of the electrodes (open circuit) (Supplementary Fig. 3a) within the interconnected vase, confirming that the photodetecting domain is continuously operational throughout its entire structure. This analysis also signifies that device

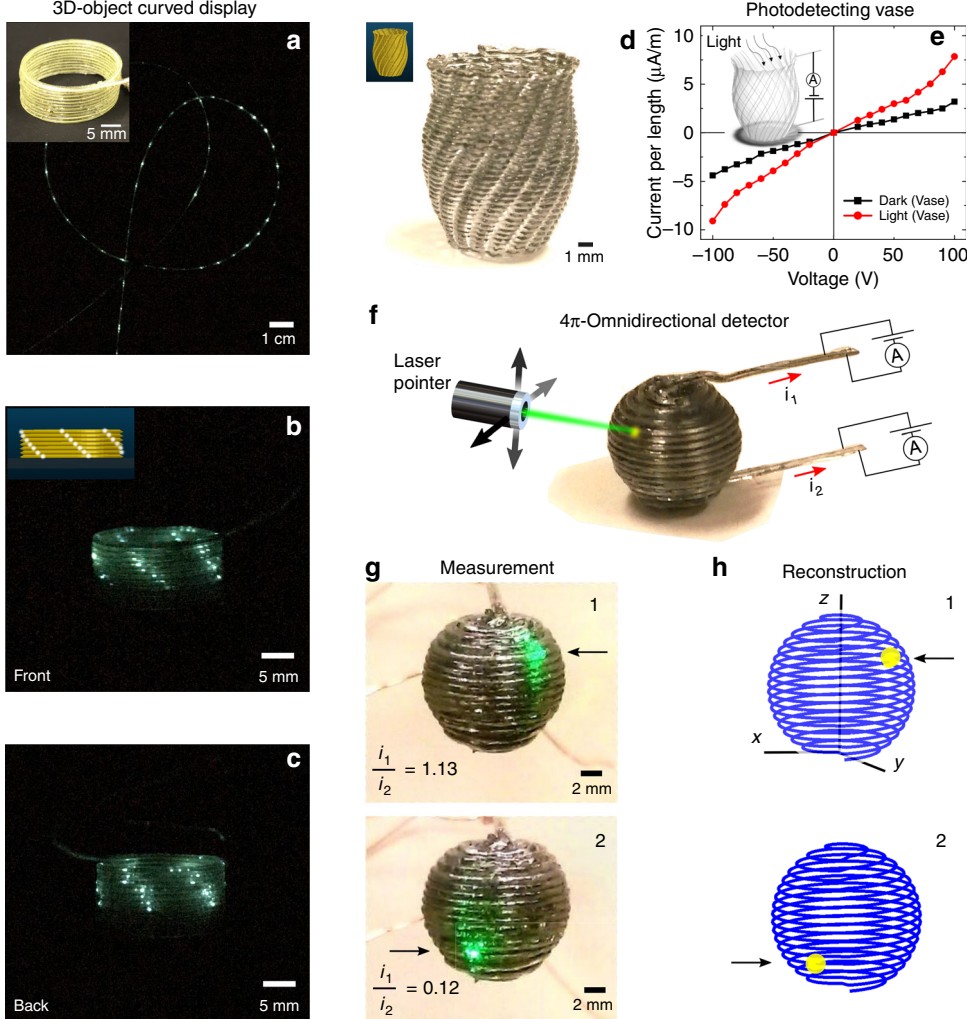

**Fig. 4** Three-dimensional printed displays and sensors. **a** Photograph of a filament dotted with 90 pixelated light emitters. The inset shows the cylinder printed from this filament, which is capable of **b**, **c** displaying electrically-activated stripe patterns all around its body. The inset of **b** shows the desired light design. **d** A patterned vase, designed as stacked layers of serpentines, is printed from the light-detecting filament and is capable of **e** detecting light and producing photocurrent from its entire structure when impinged with light. The design file for the patterned vase is from Hakalan at Scripted Vases (https://www.thingiverse.com/thing:104694) under the CC BY 3.0 license: https://creativecommons.org/licenses/by/3.0/. **f** A printed sphere with the ability for omnidirectional localized-sensing anywhere on its surface. To test its detecting accuracy, a low-power laser pointer shines at **g** different points (1 and 2) on the sphere, producing distinct current ratios ($i_1/i_2$) which allows for **h** exact imaging and reconstruction of the laser spots. All connections for the light-detecting macrostructures are made across opposite CPE electrodes

functionality can be embodied in numerous high-curvature serpentines (curvature radius of 0.38 mm) of the multilayered patterned vase (Supplementary Fig. 6). In addition, with no need for additional steps such as pick-and-place of external discrete sensors or curing of interconnects[31,32], coupled with our print approach being a continuous-flow process, complex macrostructures such as the printed patterned vase (3.3 cm³) can be formed quickly (10 min) with fully-connected microscale devices integrated seamlessly throughout its centimeter-scale body.

So far, sensors[12,14] printed from conductive inks are planar in their architectures. Sensors embodied in a three-dimensional structure can, however, offer a higher dimensional degree of sensing as compared to flat 2D-sensor. Harnessing this ability to form customizable-shapes with full-body sensing capability, we printed a closed sphere (Fig. 4f) capable of 4π-steradian omnidirectional localized light-sensing anywhere on its surface. Local detection is facilitated by the voltage-graded potential across the length of the filament[42] where it is found that one can derive the exact position of a light spot on a filament based on its

photocurrent feedbacks (Supplementary Fig. 7). We tested the omnidirectional localized light-detecting ability of the printed sphere by impinging laser light at 3 different positions (Fig. 4g and Supplementary Fig. 7c). One can observe that there is a close match between these actual impingement positions to the reconstructed positions, highlighting the accuracy in spatially detecting small light spots (0.12 mm²) from the large printed sphere (~310 mm²).

Finally, to illustrate the utility of our capability in forming complex freeform three-dimensional devices, we combine the functionalities of both light-emitting and light-detecting within a single printed structure by printing them into an aeroplane wing (Fig. 5a, b) which is capable of detecting structural defect at any point within the wing. The functional structure of the wing is described in Fig. 5c, which shows light-emitters at the top and bottom layers while light-detectors are printed within the bulk of the wing. The operation of defect localization is as follows: As the light-emitters are operated, the photodetectors generate a photocurrent, with its magnitude corresponding to the length

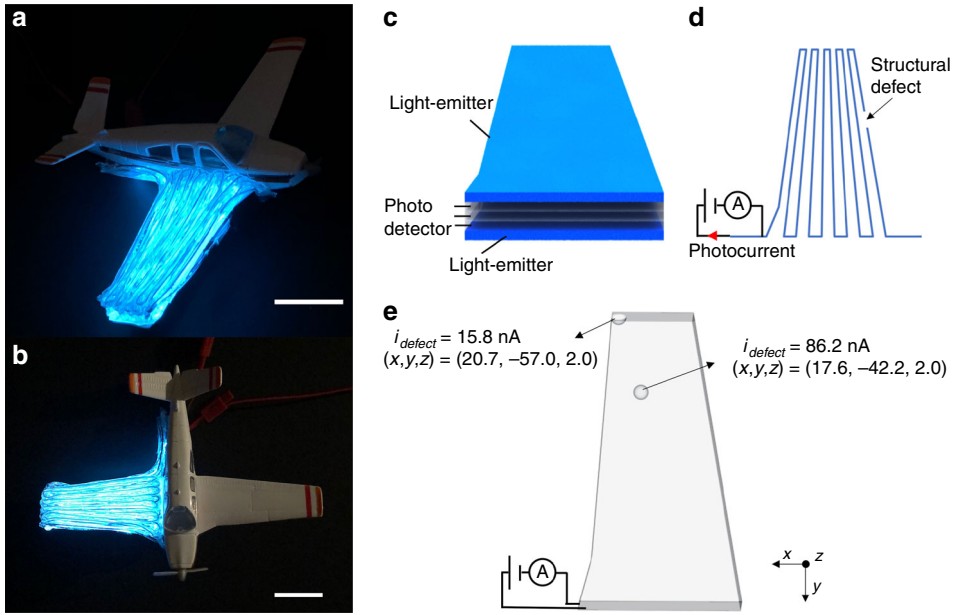

**Fig. 5** Application of a printed bi-functional light-detecting and light-emitting 3D-structure in detecting structural defects. **a** Photograph of a printed aeroplane wing that has light-emitters at the top and bottom layers, and, light-detectors in the bulk of the wing. Scale bar, 2 cm. **b** Top-view photograph of the printed aeroplane wing. Scale bar, 2 cm. **c** Schematic of the print layers of the aeroplane wing. Top and bottom layers are light-emitters, and the middle 3 layers are light-detectors. **d** Schematic of the print path for 1 layer. A photocurrent is measured from an external multimeter connected to the opposing electrodes within the printed photodetecting filament. The presence of a structural defect reduces the magnitude of the photocurrent. **e** Spatial detection of structural defects at 2 points within the wing by measuring the magnitude of the photocurrent after defects are made. The spatial coordinate of the defect is obtained by correlating the photocurrent magnitude with the severed length of the photodetecting filament

of the printed photodetecting filament. Upon the occurrence of a structural defect to the printed structure, the length is cut short, reducing the photocurrent. Through the relationship between the measured photocurrent and length of the printed photodetecting filament (See Methods for the relevant equation), one can thus spatially detect the localized position of damage within the structure (Fig. 5d, e). This capability to interrogate structural flaws is highly applicable for objects that are prone to collisions and high mechanical stresses such as 3D-printed drones, prosthetics and parts of mobile vehicles, in which the lack of ability to uncover internal structural defects can lead to failures[43,44]. In addition, as a figure of merit of the advantages of our approach, we compare the volumetric speed in forming 3D devices between our print approach and other 3D device fabrication methods. Taking the example of this printed wing, fully-connected device elements are incorporated throughout its ~ 4700 mm³ structure and were printed in a time of 24 min. The calculated volumetric speed for fabricating this 3D device wing is ~200 mm³/min which is one to three orders of magnitude faster than existing methods such as embedded-3D printing[32] (~5.1 mm³/min), kirigami[45] (~32.4 mm³/min), and manual assembly[46] (~0.2 mm³/min). (See Supplementary Table 1 that details the comparison between the different 3D device fabrication approaches). Finally, we note that the fabrication of such highly-complex device wing shape can only be achieved through our volumetric freeform device-print approach that allows for full shape customizability.

## Discussions

The structures produced from our multiscale print approach present several advantages. First, light can be locally detected or emitted at high resolution from an arbitrary 3D macro structure from simply two connections made at both ends of the printed filament, as opposed to the cumbersome N connections required for N discrete devices. Second, unlike planar sensors, fully-closed

structures like the presented printed sphere is uniquely capable of full omnidirectional sensing of light without additional bulky optical components such as mirrors or lenses, reducing the total footprint spanned by this printed optical system. Such customized omnidirectional light-sensing structures also have particular applications in solar tracking for satellites and energy harvesting, light management, or artificial eyes for robotics. In addition, our print approach is applicable to a broad set of designable microstructured multimaterial filaments, in which the filament core can combine disparate material classes of different topologies and geometries to create varying functionalities with the only material requirement that the cladding material is viscoelastic. We also note that the material selection for the filament core is dependent not only on its material properties, but also its dimensions and the required print geometry. For instance, highly brittle materials with large core diameter are printable in low curvature turns but they are not able to withstand the high bending strains present in high curvature turns. As an example, silica glass with a core diameter above 300 μm fractures and cracks if printed to a curvature radius of 7.5 mm. However, if the required radius of curvature of the print geometry is more than 7.5 mm or if the core diameter is smaller than 300 μm, the silica filament can be printed without cracking (See Supplementary Figs. 8 and 9 for the limits on the core diameter and curvature radius). In addition, we note that the photosensitive material $As_2Se_5$ used in this work has lower responsivity and bandwidth as compared to well-established materials, such as GaAs, Ge, and Si. As such, part of our future work includes printing efficient GaAs- and Si-based photodiode fibers[47], which are similarly cladded by the printable viscoelastic polycarbonate used in this work. By printing these photodiode fibers through our approach, we foresee that efficient low-powered electronic systems can be made beyond the existing planar "2D" architecture and into complex three-dimensional shapes. Using our approach, it is also envisioned that sensing and feedback can be mapped and displayed

with finer resolution in three dimensions, which is also ideal for analogous systems that require spatial sensing across its three-dimensional structure such as haptic robotics[48] and health-monitoring[49] implants. These results set the foundation for the scalable formation of 3-dimensional multiscale objects, that can be customized in both its structural geometries and functionalities, enabling a variety of device-printing applications.

## Methods

**Preparation of light-emitting filament preform**. A hollow cylindrical polycarbonate rod (McMaster-Carr) with diameter of 25 mm is prepared. The first hollow channel, to be fed with the phosphor-coated wire, is drilled at a position, 1.5 mm to the right of the center of the rod's cross-section, and has a diameter of 9.3 mm. This channel is drilled totally through the length of the preform. The second hollow channel, meant for the BiSn, is drilled at a position, 4.3 mm to the left of the center of the rod's cross-section, and has a diameter of 1.5 mm. This channel is drilled to half the length of the preform. The third hollow channel, to be fed with the W electrode wire, is drilled at a position, 6.25 mm to the left of the center of the rod's cross-section, and has a diameter of 1.5 mm. This channel is drilled totally through the length of the preform. BiSn powder is filled into the second hollow channel. The preform with the BiSn powder is then placed vertically in a vacuum oven with temperature of 150 °C for 1.5 h. The BiSn powder will melt and flow to fill up the hollow channel, hence forming a cylindrical BiSn core. COC (Grade 8007, Topas) thin films are used to rolled up the channel until the preform reached a final diameter of 35 mm. This COC-coated preform is later consolidated in the oven at a temperature of 120 °C for 1 h.

**Draw process of light-emitting filament**. The preform is first attached to a hollow preform holder. The phosphor-coated wire (purchased from KPT (Keyan Phosphor Technology) Shanghai) and W electrode wire (Goodfellow) runs through both the hollow holder and the hollow channels in the preform. The diameter of the phosphor-coated wire is 0.27 mm, and the diameter of the W electrode wire is 0.05 mm. The spools containing the phosphor-coated wire and W electrode wire are later attached to spool holders. During the thermal draw process, the temperatures of the draw tower furnace are set to be 150 °C for the top zone, 270 °C for the middle zone and 110 °C for bottom zone for bait-off, and the middle zone temperature is slowly reduced as time progresses. As the diameter of the preform is being narrowed down to a small fiber diameter, the size of the hollow channels will be reduced too. When the hollow channel decreased to a size equivalent to the diameter of the phosphor-coated wire (or the electrode wire), the channel which was once hollow is now being filled by the phosphor-coated wire (or the electrode wire). At the same time, the walls of the hollow channel are now able to catch onto the phosphor-coated wire (or the electrode wire). Hence, as the fiber is being pulled downwards by the capstan, the wires are also being pulled downwards into the capstan, allowing for unwinding of wires from their spools. The wires can then be continuously fed into the preform as the fiber drawing proceeds.

**Final light-emitting filament**. The filament consists of a multimaterial functional domain (FD) that comprises of an active light-emitting wire that has a 0.20 mm-diameter inner copper electrode, layered by an 18 μm-thick dielectric and lastly capped by an 18 μm-thick electroluminescent (EL) ZnS-phosphor outer layer. A conductive tungsten electrode (diameter of 50 μm) is spaced 0.05–0.1 mm away from the ZnS, and a 40 μm-diameter conductive BiSn cylindrical core is centrally situated with no connection to tungsten and ZnS. The 0.5–0.7 mm diameter material polycarbonate (PC) holds all of these materials and interfaces in position. Lastly, a 0.2–0.4 mm thick cyclic olefin copolymer (COC), which envelopes the cylindrical polycarbonate core, serves as the printable thermoplastic.

**Laser-induced capillary breakup**. The laser used here has a wavelength of 808 nm, spot size of 50 μm, operating at pulse mode with frequency of 5 kHz and a duty of 2%. The fiber is translated at a speed of 1.4–2.8 μm/s relative to the laser. This wavelength is chosen because the polycarbonate cladding is transparent to the laser. Thus, we can achieve the effect of "inside-out" heating, in which the filament is not deformed from Rayleigh instability, thus retaining its cylindrical shell necessary for the print process. Focusing the laser onto the BiSn metal core, BiSn absorbs the energy and becomes heated. The heat is being transferred to surrounding polycarbonate. The temperature of the surrounding polycarbonate increases beyond its glass transition temperature, decreasing its viscosity. The formation of flowy PC matrix helps shape the BiSn core into thermodynamically-stable spheres. Using a laser set-up also allows one to obtain spheres only at desired localized spatial positions along the length of the fiber. This can be done by translating the fiber to the position desired while the laser position is fixed. Smaller pixel-spheres can be attained by reducing the spacing between the tungsten wire and the phosphor surface. Higher pixel density can be achieved by increasing the diameter of the BiSn cylinder while keeping the spacing between the tungsten and phosphor surface constant.

**Preform making and thermal drawing parameters of light-detecting filament**. The photoconductive chalcogenide glass (amorphous $As_2Se_5$) is shaped into a cylinder by using the seal-ampoule melt-quenching, in which the sealed ampoule containing powdered $As_2Se_5$ was heated to 650 °C for 10 h in a rocking furnace to ensure a homogeneous cylindrical shape. The glass liquid was cooled to 300 °C before quenching in water. The filament of length hundreds of meters long were drawn from a macroscopic cylindrical preform of outer diameter of 35 mm which contains cylindrical chalcogenide glass $As_2Se_5$, 4 mm in diameter. The glass is contacted at opposite sides by two electrodes made up of conducting polyethylene with rectangular cross-section of width 3 mm and height of 2.6 mm, and cladded by a polycarbonate layer, which is transparent to the visible and near-infrared wavelength. The preforms were consolidated for 45 min at 190 °C under vacuum. Finally, the preform is thermally drawn in a three-zone vertical tube furnace with the top-zone temperature 150 °C, middle-zone temperature 270 °C and bottom-zone temperature 110 °C. The final diameter of all printing filament ranges from 0.6 mm to 0.8 mm.

**Final light-detecting filament**. The functional domain within the drawn photo-detecting filament used to print the pyramid is composed of a 96 μm diameter $As_2Se_5$ amorphous semiconducting core, which is contacted with two juxtaposed rectangular conducting polyethylene (CPE) electrodes, with a cross-sectional dimensions of 73 by 61 μm. To allow for larger axial non-uniformity in the voltage profile, the filament used to print the patterned vase and omnidirectional sphere has CPE with dimensions of 14 μm by 10 μm and $As_2Se_5$ core dimension of 92 μm. It is noted that a smaller CPE prevents the blockage of light from impinging onto the $As_2Se_5$ core, thus enhancing the current values measured at both ends.

**Printer and nozzle set-up**. The printer used is Rova3D multi-nozzle printer and the software used to generate and read the gcode for printing are Slic3r and Prometheus, respectively. The hot end is made up of a stainless steel tube that can range up to 2 mm in length, and is heated by a nichrome wire. A high-temperature insulation tape is inserted between the hot end and the region above the hot end, which is termed as the cold end. The cold end is ensured to have a temperature below the glass transition temperature of the filament cladding. The temperature of the cold end can be regulated by a cooling system pre-installed in the printer. A video of the print process, as well as the modified nozzle, is shown in Supplementary Video 1.

**Print parameters for light-emitting filament**. The light-emitting star and cylinder are printed with a single continuous light-emitting filament and with a print speed of 100 mm/min at a temperature of 220 °C with the ratio of the depositing speed and printing speed kept at 1:1. The 3D designs of the star and cylinder are designed and drawn in Solidworks, and processed in the software, Slic3r, to output a gcode file for printing. The gcode file is read by the Pronterface Software which communicates the xyz print coordinates to the 3D-printer. A COC layer is used at the bottom layer to increase adhesion of the first layer of the printed product to the printing bed. The temperature of the printing bed is kept from 65 °C to 75 °C. The temperature of the hot-end is measured by a VWR International thermocouple probe. The printed layer height ranges from 0.70 mm to 1.10 mm. To make connection to the external voltage supply, we shave the claddings and ZnS off at the one end of the filament, exposing the tungsten and copper wires for connections.

**Print parameters for light-detecting filament**. The photodetecting pyramid, sphere and vase are printed with a single continuous light-detecting filament. The pyramid and sphere are printed at a speed of 145 mm/min at temperature of 270 °C and the vase is printed at 175 mm/min at a temperature of 290 °C. The 3D design for the pyramid and sphere are designed and drawn in Solidworks. The 3D design file for the vase is from Hakalan at Scripted Vases (https://www.thingiverse.com/thing:104694) under the CC BY 3.0 license: https://creativecommons.org/licenses/by/3.0/. The design files of the pyramid, sphere and vase are processed in Slic3r to output their corresponding gcode print files and later read in Pronterface to send the xyz print coordinates to the printer. The ratio of the depositing speed and printing speed for both structures is kept at 1:1. A polycarbonate layer is used at the bottom layer to increase adhesion of the first layer of the printed product to the printing bed. The temperature of the printing bed is kept from 120 °C to 130 °C. The temperature of the hot-end is measured by a VWR International thermocouple probe. The printed layer height ranges from 0.45 mm to 0.65 mm. In order to determine the continuity of the $As_2Se_5$ core after printing, the printed multi-material filament is etched with Dichloroethane solution to remove the PC cladding, and the CPE is later carefully removed, which then exposes the $As_2Se_5$ core. Connections of all printed devices to the power supply and characterization tools are manually done by using a razor blade to carefully shave the outer PC cladding and exposing the CPE electrodes.

**Characterization of pixels**. The plot of width of light-emission vs AC voltage is obtained by taking optical micrographs of the lighted pixels for increasing root-mean-square AC voltage. We use a function generator to generate a sinusoidal wave and amplify its voltage (by 1000) through a Locked-in amplifier (Stanford Research Systems, Model SR830). The frequency of the sine wave used is 14 kHz and the

exposure time in capturing the optical micrographs is kept constant at 20 ms. Using these optical images, we draw a line scan across the light-spots and extract its grey-scale intensity line profile in the software ImageJ. Later, the profile of each line-san can be Gaussian curve-fitted using the software IgorPro. We then extract its standard deviation ($\sigma$). Here, we equate the width of light-emission to be $2\sigma$.

**Lighting-up the cylinder display**. The connections were made across the copper within the ZnS-coated wire and the tungsten wire. Similarly, the frequency of the sine wave used is 14 kHz and the RMS AC voltage applied is 250 V. In order to determine the axial position of the pixel-sphere along the fiber, we map the $(X, Y, Z)$ coordinates of the desired locations of light-ups in a 3D structure to the axial position along the fiber. (Supplementary Fig. 7b).

**IV measurements**. A 500 mW/cm$^2$ broadband white light source was used to illuminate the patterned vase. To obtain the presented IV curve, the voltage is tuned and the current collected from a Keithley Picoammeter 6487.

**Method of imaging**. The laser used for localized detection has a wavelength of 570 nm, and power of 5 mW. By using a single continuous filament as the printing ink to form 3D structures, we wrote an algorithm that maps one-to-one between the axial length of the filament and the $xyz$ coordinate of any arbitrary 3D structure (Supplementary Fig. 7a and b). Local detection from a 3D structure is enabled by a previous work[42] where it is found that the application of an electric field across the opposite ends of the polymeric electrodes (CPE) creates a convex voltage potential profile across the filament axis. A 6 cm focal length lens is used to focus the laser beam spot to ensure that the light spot falls within a single layer of the printed sphere. A voltage of 100 V is applied across the ends of the printed sphere in order to collect the photocurrents. Next, we measure the ratio of two current values collected from each end of the printed filament upon light impingement at any arbitrary position on the sphere. This ratio is distinct to the position of light impingement along the printed filament axis given by the following equation:

$$(X, Y, Z) \equiv L_n = a\left[\frac{\delta}{2}ln\left(\frac{e^{\frac{L}{\delta}} + \frac{i_1}{i_2}}{e^{-\frac{L}{\delta}} + \frac{i_1}{i_2}}\right)\right] + b \tag{1}$$

where $L_n$ is the axial position of the filament, $\delta$ and $L$ are defined as the characteristic length (85 cm) and the total length of the filament, respectively, and, $i_1/i_2$ is the measured current. $a$ and $b$ are scaling factors that are determined to be $-0.641$ and 507.19, respectively, by impinging the laser spot at 2 points along the straight printed filament sections to the spheres (Supplementary Fig. 7a) and later inserting the measured $i_1/i_2$ values with their known light positions $L_n$ into the above equation. This monotonous equation can be visualized graphically in Supplementary Fig. 7d. With the calculated value of $L_n$ and the one-to-map mapping algorithm, we can then output the $(X, Y, Z)$ coordinate position of the detected light impingement on a customizable 3D structure.

**Limitations and challenges of localized detection from a 3D structure**. The first challenge faced is in ensuring that the light spot impinges onto a single layer of the 3D structure. In order to reduce the spot size, we used a focusing lens of focal length 6 cm. The omnidirectional sphere used to detect the light spot has to be placed at a distance approximately 6 cm from the lens. However, even if the width of the light spot is smaller than the layer height of the 3D structure, there is still a probability that the light spot may impinged onto two adjacent layers. In order to overcome this, there are two approaches. First, a straightforward way is to visually ensure that the light spot falls onto a single layer itself. Second, even if light does impinge between two layers, the current collected from both ends should be of a much lower value since the region between the centers of the adjacent layers is simply the transparent polycarbonate, i.e. if the arsenic selenide core is small enough, the light spot that is impinging between the layers has a lower probability of impinging onto the arsenic selenide cores, thus the detected current will be negligible. Alternatively, as a future work to detect bigger light spot that is impinged onto 2 or more layers, one may include additional photodetecting structures (30), such as that of different characteristic length $\delta$, in order to obtain more current feedbacks upon light illumination. One can obtain an additional information, such as the width of the beam or the positions of multiple light spots, for every additional photocurrent feedback.The second challenge faced is the scattering of the light. Since it is essential for the light spot to impinge only onto a single spot on the omnidirectional sphere, an anti-scattering foam is inserted into the sphere. It is also noted that with higher intensity of the laser spot, low-intensity scattered and stray light then plays a smaller role in affecting the detected localized position. In our experiment, we show that a focused spot with full-width half maximum (FWHM) of ~0.35 mm from a regular laser pointer with power of 5 mW has intensity high enough to prevent scattering effects from affecting the localized detection.

**Printing of aeroplane wing and measurement of localized structural defects**. The aeroplane wing is printed with the light-emitting filament at the top and bottom layers, and with 3 layers of photodetecting filaments between the printed light-emitters. The 3D dimensions of the aeroplane wing is obtained by manually measuring the dimensions of the toy model plane wing (Miniature Aircraft Training Aid Bonanza from Sporty's Pilot Shop). These dimensions are later

compiled in an excel sheet to output a design file with the specific xyz coordinates of the wing. The coordinates of the design and its corresponding print speeds are then combined into a text file. This text file is send to the Pronterface Software which communicates the xyz coordinates for printing. Both filaments are cladded with COC and are printed at a speed of 150 mm/min at 250 °C. The measurement of photocurrent from the photodetecting filament is made through a Keithley Multimeter by applying a voltage of 500 V. The measurement of the spatial position of localized defect makes use of the equation in ref. [42], which correlates the photocurrent to the length of the photodetecting filament. By taking the ratio of the photocurrent between that measured from the length after cutting (defect-forming) the printed filament and that measured from a reference length, we remove the dependence of the light intensity to obtain the following equation:

$$i_{\text{defect}} = \frac{\sinh\left(\frac{L_{\text{defect}}}{2\delta}\right)}{\sinh\left(\frac{L_{\text{reference}}}{2\delta}\right)} \cdot i_{\text{reference}} = A \cdot \sinh\left(\frac{L_{\text{defect}}}{2\delta}\right) \tag{2}$$

where $\delta$ (characteristic length) is calculated to be 33.98 cm, and $A$ is measured to be 0.353. By measuring $i_{\text{defect}}$, one can thus determine $L_{\text{defect}}$ which in turn corresponds to a specific xyz position within the aeroplane wing.

## Data availability
The data that support the findings of this study are available from the corresponding author upon reasonable request.

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

## Acknowledgements

G.L.Z.J., R.Y., M.R., T.K. and Y.F. were supported in part by the National Science Foundation under the Center for Materials Science and Engineering (DMR-0819762 and DMR-1419807) as well as the US Army Research Laboratory and the US Army Research Office through the Institute for Soldier nanotechnologies under contract number W911NF-13-D-0001. The authors are grateful to Dr. Eric Wetzel of the U.S. Army Research Laboratory for providing early input that was inspirational to the multimaterial printing approach, as well as strong feedback on the experiments and manuscript.

## Author contributions

G.L. and Y.F. conceived the idea of printing designable microstructured filaments, and designed the techniques and experiments to do so. G.L. and R.Y. setup the hardware and the software of 3D printing printer, and assisted in the preparation of the figures. G.L., T.K. and M.R. prepared the filaments for printing, and characterize them. G.L. and Y.J. discussed and wrote the Matlab algorithm on imaging from a 3D structure. G.L. and Y.F. wrote the manuscript, and G.L., R.Y., M.R., T.K., Y.J., J.J. and Y.F. analyzed the results, and feedback on the manuscript.

## Additional information

**Competing interests:** The authors declare no competing interests.

