## [Peer Review File · Nature Communications]

Reviewers' comments:

Reviewer #1 (Remarks to the Author):

This manuscript reports an interesting and very advanced 3D-printing technique that is capable of simultaneously printing metals, semiconductors and polymers into structured electronic devices. As this technique is a one-step process, it avoids issues causing device quality degradation, such as delamination, and time-consuming post-annealing procedures associated with other 3D-printing techniques, where different materials are printed separately. In this method, the authors first fix device components (of filamentary forms) into an adhesive tube, and they then print the adhesive tube to yield device-quality 3D structures. In this way, the authors produce a wide variety of structures including those with precisely positioned light-emitting dots at 55 micron resolution, and structures with embedded omnidirectional photodetectors capable of spatially resolving light at micron resolution across its entire centimeter-scale surface. The overall approach is novel, the technique is potentially valuable for future industrial manufacturing, the results are convincing and the quality of the manuscript is high; therefore, the reviewer recommends this manuscript to be published on Nature Communications.

Nevertheless, the reviewer suggests the authors to address the following points that might help the selection of materials for other 3D-printable devices in the future:

1. What is the advantage of using As₂Se₅ as a light detection semiconductor? Is it because it is a malleable and light-sensitive material? Some comprehensive discussion on printable and non-printable materials should be included. For example, is it possible to print monocrystalline Si? What about GaN, etc. The reader should be oriented to the strengths, weaknesses and limitations.
2. Some material used in 3D printing, such as W, is brittle. In this sense, are there any requirements for the minimal radius of curvature in the printed structure, to avoid the disconnection/break of the W filaments? Mechanical testing should be included, along with computational analysis of bending/stretching strains.

Reviewer #2 (Remarks to the Author):

The manuscript describes a technique combining existing ink- and melt-extrusion based digital manufacturing approaches with novel functional inks to realize light emitting and light detecting structures in 3 dimensions. High quality rendered illustrations and basic demonstrations of functionality are presented. Some of the interesting prior art / publications on 3D device fabrication is cited.

The use of novel inks is interesting, particularly in being able to use the process to modify the ink to create useful microstructures in situ, while leveraging the robust 3D printing platforms for this purpose could be powerful. It's intriguing to see this direction being explored by the authors, who already built up a formidable arsenal of capability in melt-spinning of multimaterial and multifunctional fibers and devices based on them.

However, while the demonstrated process capability is promising, and the basic functionality is there (light emission / detection), the paper falls somewhat short of the high standards established previously by the same group in developing processes and multi-material "feedstocks" for the fabrication of useful devices. For example, missing is a clear experimental indication of utility for the capability – the 3D sensing of light impinging on a sphere was shown a long time ago by the same group on a larger scale using multi-material fibers. Such structures can be scaled to lower dimensions in a fairly straightforward manner, it seems, so why would a device developer working on 3D light detectors (?) need to master a new tool / process / material set to do the same?.. Is

there a particular roadmap for this kind of device and its performance?.. If so, what's the relevant figure of merit?

For such application-oriented demonstrations to be taken seriously, it's important to define useful figures of merit, and make comparisons to state-of-the-art. This goes both for the performance of the demonstrated devices, as well as for the processing method itself. For example, the authors note in the introduction that pick-and-place techniques are deficient, but fail to indicate clearly and quantitatively how. Industrial pick-and-place automation is quite robust and is used routinely for rapid and cheap fabrication of all manner of sophisticated circuits on many length scales. Similarly with the comparison to kirigami-based *processes* for making structures, where industrial-scale cutting can be done using a variety of robust and highly scalable technologies. The same with prior art on 3D *devices* made using other approaches, both in what's cited and what's omitted from consideration by the authors, where merit figures are clearly stated and compared.

Importantly, for a scientific or applied journal, one would also expect to see a strong analysis of how process parameters affect the resulting devices, or how the formulation of the ink is adjusted to improve performance or resolution. The only place in the paper where that's shown is Figure 2 panel d, where 3 sizes of spheres are used – more of this type of analysis could enhance the paper and the journal's readership.

Reviewers' comments are in **bold**, the authors' responses are in Roman.

Reviewer #1:

This manuscript reports an interesting and very advanced 3D-printing technique that is capable of simultaneously printing metals, semiconductors and polymers into structured electronic devices. As this technique is a one-step process, it avoids issues causing device quality degradation, such as delamination, and time-consuming post-annealing procedures associated with other 3D-printing techniques, where different materials are printed separately. In this method, the authors first fix device components (of filamentary forms) into an adhesive tube, and they then print the adhesive tube to yield device-quality 3D structures. In this way, the authors produce a wide variety of structures including those with precisely positioned light-emitting dots at 55 micron resolution, and structures with embedded omnidirectional photodetectors capable of spatially resolving light at micron resolution across its entire centimeter-scale surface. The overall approach is novel, the technique is potentially valuable for future industrial manufacturing, the results are convincing and the quality of the manuscript is high; therefore, the reviewer recommends this manuscript to be published on Nature Communications.

We thank the reviewer for the positive comments regarding the novelty and quality of this work in 3D-printing of devices.

Nevertheless, the reviewer suggests the authors to address the following points that might help the selection of materials for other 3D-printable devices in the future: 1. What is the advantage of using As₂Se₅ as a light detection semiconductor? Is it because it is a malleable and light-sensitive material?

Response:

We thank the reviewer for the comments relating to As₂Se₅. First, we would like to emphasize that the specific example that we have provided with respect to photodetectors is not meant to limit the utilization of this general method to this particular photodetecting material. Other fiber photodetectors can be used and this serves as an example. With that being said, As₂Se₅ was chosen since it is photoconducting in the visible to near infrared range¹. As₂Se₅ has similar viscosity² with other functional low melting-points materials such as the conducting polyethylene and the polycarbonate cladding (T_g=150 °C), hence allowing us to thermally co-draw them to form a flexible semiconducting filament device with a continuous detection capability along its length. We have included the following statement in the main text to highlight the advantage of As₂Se₅ :

“We made use of As₂Se₅ due to two reasons. First, As₂Se₅ glass is a semiconductor that has a bandgap which is sensitive to light across the visible range, enabling its use as active

¹ Zakery, A. & Elliott, S. R. Optical properties and applications of chalcogenide glasses: A review. *Journal of Non-Crystalline Solids* **330**, 1–12 (2003).

² Rein, M. *et al.* Self-assembled fibre optoelectronics with discrete translational symmetry. *Nat. Commun.* **7**, 12807 (2016).

material in visible-light photodetectors. Second, As₂Se₅ glass has matching glass transition and viscosity, that allows thermal co-drawability, with other low T_g materials such as the insulating polycarbonate (PC) and electrically-conductive carbon-loaded polyethylene (CPE) to form a 1-dimensional flexible photodetecting filament device with a continuous detection capability along its length.”

Some comprehensive discussion on printable and non-printable materials should be included. For example, is it possible to print monocrystalline Si? What about GaN, etc. The reader should be oriented to the strengths, weaknesses and limitations.

We thank the reviewer for the comments relating to discussion on printable and non-printable materials. Regarding non-printable materials, we note that the cladding material has to be viscoelastic, a common property in all thermoplastics. We agree that different material properties affect the printing characteristics such as the minimal achievable radius of curvature. There are, however, no fast and hard limitation on printable materials because there are many degrees of freedom with respect to not only the choice of material, but also the filament dimensions and print geometry. As an example, we study silica glass. Silica glass is a highly brittle material with a low fracture strain. With a large core diameter, it is not able to withstand the high bending strains present in high curvature turns. For instance, silica glass with a core diameter above 300 μm fractures and cracks with a curvature radius of 7.5 mm (Supplementary Figure 8). However, by tuning the print geometry such that the required radius of curvature is above 7.5 mm or by tuning the core diameter to be smaller, the silica filament can be printed without cracking (Supplementary Figure 8a and 8b). Likewise, this analysis indicates that materials, such as Si and GaN, can only be printed if its core diameter is smaller than a specific threshold or if the required radius of curvature (R) is larger than that predicted by the elastic bending strain equation³, $R = D/2\varepsilon$, where D is the core diameter and ε is the fracture strain. We have included a text on printable and non-printable materials in the Discussions section of the main text and added the analysis of the limits of the core diameter and curvature radius in Supplementary Figure 8.

In the Discussions section:

“Our print approach is applicable to a broad set of microstructured multimaterial filaments, in which the filament core can combine disparate material classes of different topologies and geometries to create varying functionalities with the only material requirement that the cladding material is viscoelastic. We also note that the material selection for the filament core is dependent not only on its material properties, but also its dimensions and the required print geometry. For instance, highly brittle materials with large core diameter are printable in low curvature turns but they are not able to withstand the high bending strains present in high curvature turns. As an example, silica glass with a core diameter above 300 μm fractures and cracks with a curvature radius of 7.5 mm. However, if the required radius of curvature of the print geometry is more than 7.5 mm or if the core diameter is smaller than 300 μm , the silica filament can be printed without cracking. (See Supplementary Fig.8 for the limits on the core diameter and curvature radius).”

2. Some material used in 3D printing, such as W, is brittle. In this sense, are there any requirements for the minimal radius of curvature in the printed structure, to avoid the

³ Kuhn, H. & Medlin, D. *ASM Handbook, Volume 8: Mechanical Testing and Evaluation, Stress-Strain Behavior in Bending*. ASM (2000).

disconnection/break of the W filaments? Mechanical testing should be included, along with computational analysis of bending/stretching strains.

Response:

We thank the reviewer for the comments on the requirements for the minimum curvature radius. With respect to the discussion about tungsten (W), we would first like to note that this material was chosen as an illustration for this work. We routinely produce and print filaments with other types of wires such as copper wires. With that in mind, we then look into finding a general relationship that determines the minimum printed radius of curvature for an arbitrary material. To do so, we note that there are two forms of bending strains – elastic and plastic.

We first look into elastic bending strain. Again, as an example, we focus on silica glass. Silica is a brittle material that fractures at its elastic limit with no plastic deformation. Its minimal radius of curvature can thus be predicted by the theoretical elastic bending strain equation³, $R = D/2\varepsilon$, where D is the core diameter and ε is the fracture strain. To determine if this relation holds true in predicting the minimal radius of curvature, we experimentally measure the minimal radius of curvature, determined by its fracture point, for a bending silica filament (Supplementary Figure 8a). Its diameter is kept constant at 300 μm -diameter and it is bent at different curvature radii. The experimental minimal radius of curvature is measured to be 7.5 mm. This value matches to that obtained from $R = D/2\varepsilon$, which gives R equals to 7.5 mm for a silica fracture strain⁴ (ε) of 0.02. Similarly through computational (COMSOL) analysis, we plotted a graph of maximal stress versus radius of curvature (Supplementary Figure 8b) of a 300 μm -diameter bending silica filament. In this plot, the minimal radius of curvature can be predicted by the relationship between stress and curvature radius, together with its known fracture strength. As the curvature radius decreases, the stress experienced increases until it reaches the fracture strength at a curvature radius of around 8.5 mm, matching closely with that of the equation. $R = D/2\varepsilon$ is thus applicable to determine the minimal printed radius of curvature for materials that fractures at its elastic limit with no plastic deformation. From this relation, the size of the filament will evidently constrain the radius of curvature achievable on the print bed. However one could decrease the filament size to enable a lower radius of curvature needed. Nothing limits this print approach to a particular filament diameter.

Next, we look into plastic bending strain. W wires are interesting in that it undergoes plastic deformation⁵ and strain hardening⁶ before fracturing, making the study on the requirements for minimal curvature radius for materials such as tungsten to be more complex³. As such, in-depth computational analysis and numerical calculations of bending W wires are beyond the scope of this paper and will thus be part of our future work. While the relationship between minimal curvature radius and filament diameter is complex for W, we can still use the simple elastic bending strain equation to provide an *upper limit* to the minimal curvature radius for materials that plastically deform during bending. As an example, we experimentally investigated the minimal curvature radius of W wire with a core diameter of 300 μm . By bending the W wire to different curvature radii, we then determine the radius at which it

⁴ Biswas, A. K., Cherif, C., Hund, R.-D., Shayed, M. A. & Hossain, M. Influence of Coatings on Tensile Properties of Glass Fiber. *Mater. Sci.* **20**, (2014).

⁵ Levin, Z. S., Srivastava, A., Foley, D. C. & Hartwig, K. T. Fracture in annealed and severely deformed tungsten. *Mater. Sci. Eng. A* **734**, 244–254 (2018).

⁶ Scapin, M., Fichera, C., Carra, F. & Peroni, L. Experimental investigation of the behaviour of tungsten and molybdenum alloys at high strain-rate and temperature. *EPJ Web Conf.* **94**, 01021 (2015).

fractures. We found out that the 300 μm -diameter W wire fractures at a small curvature radius of 0.334 mm. This value is indeed much lower than that predicted by the elastic bending strain equation³ ($\varepsilon = D/2R$, where D is the core diameter, and R is the radius of curvature), which gives a fracture bending radius of 0.882 mm for a fracture strain⁷ of 0.17 for W. We have included a text on the minimal curvature radius of W in the figure caption of Supplementary 8c.

⁷ Levin, Z. S., Srivastava, A., Foley, D. C. & Hartwig, K. T. Fracture in annealed and severely deformed tungsten. *Mater. Sci. Eng. A* **734**, 244–254 (2018).

Reviewer #2 :

The manuscript describes a technique combining existing ink- and melt-extrusion based digital manufacturing approaches with novel functional inks to realize light emitting and light detecting structures in 3 dimensions. High quality rendered illustrations and basic demonstrations of functionality are presented. Some of the interesting prior art / publications on 3D device fabrication is cited. The use of novel inks is interesting, particularly in being able to use the process to modify the ink to create useful microstructures in situ, while leveraging the robust 3D printing platforms for this purpose could be powerful. It's intriguing to see this direction being explored by the authors, who already built up a formidable arsenal of capability in melt-spinning of multimaterial and multifunctional fibers and devices based on them.

We thank the reviewer for this comment and acknowledging the fact that many other fiber devies can be printed using this approach.

However, while the demonstrated process capability is promising, and the basic functionality is there (light emission / detection), the paper falls somewhat short of the high standards established previously by the same group in developing processes and multi-material "feedstocks" for the fabrication of useful devices. For example, missing is a clear experimental indication of utility for the capability – the 3D sensing of light impinging on a sphere was shown a long time ago by the same group on a larger scale using multi-material fibers.

We thank the reviewer for this comment on the utility of this print capability. We added a new set of experimental results on a printed model aeroplane wing that has fully-connected device elements throughout its whole structure. This wing is printed totally with the light-detecting and light-emitting filaments. Due to the advantage in incorporating devices throughout the whole volume of the 3D structure, we are then able to interrogate structural defect at any point within the volume of the wing- an ability which is highly required in high-stress and collision-prone applications^{8,9} such as 3D-printed drones, planes, and prosthetics. This new result highlights the unique capability of our approach in forming fully-filled, volumetric, and complex freeform device shapes that are not attainable by other approaches. We have added this new result as Figure 5 and included a description in the main text:

“Finally, to illustrate the utility of our capability in forming complex freeform three-dimensional devices, we combine the functionalities of both light-emitting and light-detecting within a single printed structure by printing them into an aeroplane wing (Figure 5a and 5b) which is capable of detecting structural defect at any point within the wing...”

⁸ Mueller, E. M., Starnes, S., Strickland, N., Kenny, P. & Williams, C. The detection, inspection, and failure analysis of a composite wing skin defect on a tactical aircraft. *Compos. Struct.* **145**, 186–193 (2016).

⁹ Al-Fakih, E. A., Abu Osman, N. A. & Mahmud Adikan, F. R. Techniques for interface stress measurements within prosthetic sockets of transtibial amputees: A review of the past 50 years of research. *Sensors (Switzerland)* **16**, 1119 (2016).

Such structures can be scaled to lower dimensions in a fairly straightforward manner, it seems, so why would a device developer working on 3D light detectors (?) need to master a new tool / process / material set to do the same?.. Is there a particular roadmap for this kind of device and its performance?.. If so, what's the relevant figure of merit?

Response:

We thank the reviewer for the comments on the advantages over previously reported works, in specific Abouraddy *et al.*¹⁰ We note that manual assembly of fibers as reported in Abouraddy *et al.* and 3D printing of devices reported in this work are two distinct levels of fabrication. The former showcases simple macroscale shapes with low spatial precision at the scale of centimetres, while our work on 3D printing of devices achieves microscale to sub-mm precision with the ability to construct a variety of sizes and programmed complex architectures that are beyond simple structures such as a sphere. For example, in this work, we have printed highly complex architectures, such a vase made up of numerous high curvature serpentine turns (curvature radius ~ 0.38 mm) and an aeroplane wing (new results highlighted in Figure 5), which cannot be achieved simply by manually assembling individual fibers. This is because to form such complex shapes, it is necessary for the fibers or filaments to first thermally deform and fuse to each other, and second, be deposited and placed at high spatial precision — both of which can only be achieved through our approach. To summarize these findings, we have included a table (Supplementary table 1), detailing our advantages in fabrication precision and customizability in shapes as compared to manual assembly mentioned in this reference.

For such application-oriented demonstrations to be taken seriously, it's important to define useful figures of merit, and make comparisons to state-of-the-art. This goes both for the performance of the demonstrated devices,

We thank the reviewer for the comments in regards to the comparison in the performance of our printed photodetecting device to those in literature. We note that the chalcogenide photosensitive material As_2Se_5 used in this work has lower responsivity and bandwidth as compared to well-established materials such as GaAs, Si, or Ge that are used in embedded 3D-printing. Hence, as part of our future work, we will be printing our recently-reported efficient photodiode fibers¹¹ that contains materials such as GaAs and Si. Importantly, these fibers are cladded by the same polycarbonate filament cladding reported in this work, hence these fibers are printable through our approach. We have included a text in the Discussions section that highlight this limitation and our future work in printing highly efficient diode fibers.

In the discussion section:

“In addition, we note that the photosensitive material As_2Se_5 used in this work has lower responsivity and bandwidth as compared to well-established materials such as GaAs, Ge and Si. As such, part of our future work includes printing efficient GaAs- and Si-based photodiode fibers⁴⁵, which are similarly cladded by the printable viscoelastic polycarbonate used in this work.”

¹⁰ Abouraddy, A. F. *et al.* Large-scale optical-field measurements with geometric fibre constructs. *Nat. Mater.* **5**, 532–536 (2006).

¹¹ Rein, M. *et al.* Diode fibres for fabric-based optical communications. *Nature* **560**, 214–218 (2018).

For such application-oriented demonstrations to be taken seriously, it's important to define useful figures of merit, and make comparisons to state-of-the-art. This goes both for the performance of the demonstrated devices, as well as for the processing method itself. For example, the authors note in the introduction that pick-and-place techniques are deficient, but fail to indicate clearly and quantitatively how. Industrial pick-and-place automation is quite robust and is used routinely for rapid and cheap fabrication of all manner of sophisticated circuits on many length scales.

Response:

We thank the reviewer for the comments on the figure of merits of our print approach as compared to previous 3D device fabrication approaches. In this revised manuscript, the key figure-of-merit that differentiates our work over other 3D-device fabrication approaches is the *volumetric speed* of fabricating the 3D devices. We note that *volumetric speed* depends on the *total device volume* divided by the *time for fabrication*.

We agree that industrial pick-and-place is fast, but they are optimized for flat planes. Pick-and-place of devices into 3D-printed objects, or commonly known as Embedded 3D-printing, are still done manually because 3D objects are non-planar. For example, see Ref. 12 which shows the embedding of devices into a cube. In addition, even if the embedding speed is increased, the bulk of the time in fabricating the 3D-device is spent on curing the conductive inks which takes hours per printed layer.^{12,13,14} This is unlike our approach which does not need any post-print processes such as curing since all connectorizations are already pre-made within the filament itself.

Similarly with the comparison to kirigami-based *processes* for making structures, where industrial-scale cutting can be done using a variety of robust and highly scalable technologies. The same with prior art on 3D *devices* made using other approaches, both in what's cited and what's omitted from consideration by the authors, where merit figures are clearly stated and compared.

We note that kirigami and 3D-printing of devices are two different forms of fabrication with each having its own advantages. Kirigami-based processes has the advantage of producing thin 3D-like device structures that can be foldable, stretchable and flexible through cuts. We also agree that the speed of laser cutting is fast for the fabrication of 3D-device in kirigami approaches. However, while the area of kirigami devices is large, its thickness has to be thin to facilitate folding and to ensure that the device can be laser-cut.¹⁵ As such, the total device volume (area multiplied by thickness) is small, leading to a lower volumetric speed of 3D-device fabrication. Moreover, due to its pop-out 2D nature, kirigami structures have device functionality only at the surface of the three-dimensional structure and not within its bulk, while our 3D-printing approach can enable fully-filled volumetric 3D device structures as

¹² MacDonald, E. *et al.* 3D printing for the rapid prototyping of structural electronics. *IEEE Access* **2**, 234–242 (2014).

¹³ Li, J. *et al.* Hybrid additive manufacturing of 3D electronic systems. *J. Micromechanics Microengineering* **26**, (2016).

¹⁴ Lopes, A. J., MacDonald, E. & Wicker, R. B. Integrating stereolithography and direct print technologies for 3D structural electronics fabrication. *Rapid Prototyp. J.* **18**, 129–143 (2012).

¹⁵ Lamoureux, A., Lee, K., Shlian, M., Forrest, S. R. & Shtein, M. Dynamic kirigami structures for integrated solar tracking. *Nat. Commun.* **6**, (2015).

represented by our printed densely-filled plane wing device (Figure 5). In addition, the shapes of kirigami devices are restricted by design rules on folding configurations, making it challenging to form complex structures such as a printed vase with customized structural features or a printed aeroplane wing as shown in this work. To summarize all of these comparisons, we have included a table (Supplementary Table 1) that further elaborates their characteristics, in relation to the time, device volume, speed and shapes, of the different 3D-device fabrication methods.

Finally, we make comparisons of the merit figure, volumetric speed, of our print approach with other fabrication methods. Taking the example of the printed model aeroplane wing, fully-connected device elements are incorporated throughout its $\sim 4700 \text{ mm}^3$ structure and were printed in a time of 24 minutes. This gives a speed of $\sim 200 \text{ mm}^3/\text{min}$, which is around one to three orders of magnitude faster than existing 3D-device fabrication methods such as embedded 3D-printing¹³ ($\sim 5.1 \text{ mm}^3/\text{min}$), kirigami¹⁵ ($\sim 32.4 \text{ mm}^3/\text{min}$) and manual assembly¹⁶ ($\sim 0.2 \text{ mm}^3/\text{min}$). Moreover, the fabrication of such highly-complex device wing shape can only be achieved through our freeform print approach. We have included these discussions and added the references of the kirigami and manual assembly approaches in the main text:

“Taking the example of this printed wing, fully-connected device elements are incorporated throughout its $\sim 4700 \text{ mm}^3$ structure and were printed in a time of 24 minutes. The calculated volumetric speed for fabricating this 3D device wing is $\sim 200 \text{ mm}^3/\text{min}$ which is one to three orders of magnitude faster than existing methods such as embedded-3D printing³² ($\sim 5.1 \text{ mm}^3/\text{min}$), kirigami⁴⁵ ($\sim 32.4 \text{ mm}^3/\text{min}$), and manual assembly⁴⁶ ($\sim 0.2 \text{ mm}^3/\text{min}$). (See Supplementary Table 1 that details the comparison of volumetric speed of different 3D device fabrication approaches). Finally, we note that the fabrication of such highly-complex device wing shape can only be achieved through our volumetric freeform device-print approach that allows for full shape customizability. “

Importantly, for a scientific or applied journal, one would also expect to see a strong analysis of how process parameters affect the resulting devices, or how the formulation of the ink is adjusted to improve performance or resolution. The only place in the paper where that's shown is Figure 2 panel d, where 3 sizes of spheres are used – more of this type of analysis could enhance the paper and the journal's readership.

Response:

We thank the reviewer for the comment on the addition of how process parameters and ink formulation can be used to improve performance. In this revised manuscript, we have included plots of “Temperature vs print speed” for filaments of different polymeric claddings in Figure 3. Within these plots, different coloured sections depict the outcomes of the printed devices for different speeds and temperatures. In the main text, we have also included additional information in the main text on how engineering the cladding materials can increase the adhesion of the print layers while ensuring that the device functionality is retained and incorporated in the 3D-structure.

¹⁶ Miller, A. J. *et al.* Compact cryogenic self-aligning fiber-to-detector coupling with losses below one percent. *Opt. Express* **19**, 9102 (2011).

In the main text:

“We later show in Figure 3e and 3f, the ideal process parameters, namely the print speed and nozzle temperature, required to achieve both good polymeric adhesion and retainment of device functionality for the light-detecting and light-emitting filaments. Comparing the parameter map between both filaments, we note that the inclusion of a lower T_g outer adhesion cladding around a higher T_g core expands the speed-temperature area in which one achieves the desired strong adhesion with preservation of functionality. This strategy of elevating the glass transition temperature from the outer filament cladding to the inner core material reduces the failure of filament device printing, and can thus be implemented for future printing of multimaterial filaments with functionalities other than optoelectronics as described in the work.”

Reviewer #3 :

The authors demonstrate Simultaneous 3D printing of different materials for creating electronic devices. They show an approach that allows for unprecedented control on the spatial localization of each of the different materials inside one single filament. This control is what enables the different ink functionalities. Most of previous works have shown that it is possible to 3D print multi-materials, but only PASSIVE structures. This work's novelty resides in that it enables actual device functionalities on 3D printed structures. All of this is done with a single filament and a single printing head. And most importantly: it's scalable. The micro-structured filaments are kilometers long with printing speeds of up to 145mm/min. In order to ensure that the core functionality stays unchanged during the printing process while enabling printing of a range of materials, the authors use a nozzle with temperature below the glass transition of the filament core (where the multiple materials are embedded) and above the glass transition of the Polycarbonate cladding (which serves as the print medium and adhesion layer).

We thank the reviewer for recognizing and acknowledging the novelty and scalability of our approach in enabling device functionalities in 3D printed structures.

The work is extremely relevant to a wide range of applications and likely to have a very high impact to the broader scientific community. The simplicity of the proposed filament fabrication method (filaments were fabricated using thermal drawing of a multimaterial preform on a draw tower furnace) and the immediate practicality of the demonstrated applications can open the door for creative and increasingly complex combinations of materials for novel applications. The approach works with any Fused Deposition Modeling 3D printer with a minimum modification of the nozzle. These kind of 3D printers are sold and available everywhere from universities and companies to the houses of hobbyists (the Columbia MakerSpace has like 12 of them). Availability of these functional filaments would definitely unlock several innovations.

We thank the reviewer for the positive comments on the high-level practicality of our work which is applicable for the scientific community, industries and universities alike.

Minor points that should be addressed:

- How would the authors address the compatibility with existing “2D” electronic systems (which are usually driven by few volts)?**

We thank the reviewer for the comment on the compatibility with existing planar electronic systems. We would first like to highlight that our print approach is applicable and compatible in printing high-temperature materials present in existing low-power planar electronic systems. For example, in this work, we have printed materials such as copper and tungsten in the form of wires incorporated in the light-emitting filament. In addition, as part of our future work, we will be printing our recently-reported photodiode fibers¹⁷ that contains active device materials such as gallium-arsenide (GaAs) and silicon (Si). These fibers contain embedded commercial-grade devices present in existing electronic systems that are both efficient and low-power. Importantly, they are cladded by the same polycarbonate filament cladding reported in this work, hence these fibers are printable through our approach. By

¹⁷ Rein, M. *et al.* Diode fibres for fabric-based optical communications. *Nature* **560**, 214–218 (2018).

printing these photodiode fibers through our approach, efficient low-powered electronic systems can now be made beyond the existing planar “2D” architecture and into complex three-dimensional shapes. We have added a description of this response in the Discussion section of the main text:

In the Discussion section:

“As such, part of our future work includes printing efficient GaAs- and Si-based photodiode fibers⁴⁷, which are similarly cladded by the printable viscoelastic polycarbonate used in this work. By printing these photodiode fibers through our approach, we foresee that efficient low-powered electronic systems can be made beyond the existing planar “2D” architecture and into complex three-dimensional shapes.”

• How far down the filament’s diameter can be pushed? The paper demonstrates 0.5mm print precision and mentions that higher printing precision can be achieved using nozzle hot ends of smaller diameters. The placement of the light emitters inside the filament is accurate to hundreds of microns along the filament (each emitter sphere has a diameter of 55um).

We thank the reviewer for the comment on the minimal printable filament diameter for maximal print precision. We have added our analysis on the minimal printable filament diameter in Supplementary Note 2. In this analysis, we first highlight that the factor, impeding a filament diameter smaller than 0.5 mm to be printed, is attributed to filament buckling^{18,19}. Filament buckling, which leads to jamming and congestion of the heated nozzle, arises in specific from our printer infrastructure. In our printer infrastructure, feed rollers are used to push the filament through the heated nozzle. As the rollers push the filament, the filament is axially compressed along its length, leading to buckling especially as the filament diameter is smaller. To determine the smallest achievable fiber diameter, we then use the Euler Buckling equation¹⁸:

$$\sigma_{cr} = \frac{\pi^2 E}{16 \left(\frac{L}{D}\right)^2 K}$$

, where σ_{cr} is the critical stress at which the filament buckles, E is the elastic modulus of the filament, L is the length between the feed rollers and the heated nozzle, D is the filament diameter and K is the corrective factor as the nozzle diameter is slightly bigger than the filament diameter. Through some re-arrangement of the above equation, one can then correlate the filament diameter, D, to the length, L:

$$D = \sqrt{\frac{16 K \sigma_{cr}}{\pi^2 E}} \cdot L = A \cdot L$$

, where A is a proportionality constant which equates to 8.33×10^{-4} by substituting $D = 0.5$ mm and $L = 600$ mm measured for our printer setup. From this equation, one can observe that

¹⁸ Yang, Z., Jin, L., Yan, Y. & Mei, Y. Filament breakage monitoring in fused deposition modeling using acoustic emission technique. *Sensors (Switzerland)* **18**, (2018).

¹⁹ Venkataraman, N. *et al.* Mechanical and rheological properties of feedstock material for fused deposition of ceramics and metals (FDC and FDMet) and their relationship to process performance. *Proc. Solid Free. Fabr. Symp.* 351–359 (1999).

the minimum printable filament diameter is dependent on the length, L, between the rollers and the nozzle. As such, by decreasing L through modifying the rollers to be closer to the nozzle, the probability of buckling decreases, allowing for filament diameter much smaller than 0.5 mm to be printed. For example, by decreasing L to a reasonable 10 cm, one could in principle print ~ 80 μ m-diameter filaments, matching close in dimensions with the size of the emitter spheres. We have also added a line in the main text to direct readers to Supplementary Note 2 to inform them about our analysis above and the approaches towards improving the print precision.

In Main Text:

“Higher print precision (up to 0.5 mm shown in this work) can easily be achieved by using thermally-drawn filaments and hot ends of smaller diameters (See Supplementary Note 2 on the approach and the factors to improve on the print precision).”

In Supplementary Note 2:

“In the main text, we describe a print precision of 0.5 mm. To increase the print precision of this device-printing approach, both the diameter of the filament and the nozzle orifice have to decrease. The size of the nozzle orifice can be reduced through precise machining methods such as laser cutting or milling, while the filament diameter can be reduced by increasing the draw speed during thermal drawing of the filament. However, the main contributing factor that impedes a filament diameter smaller than 0.5 mm to be printed in this work, is attributed to filament buckling^{18,19} ...”

REVIEWERS' COMMENTS:

Reviewer #1 (Remarks to the Author):

The authors have provided complete responses to all of the referee comments. I feel that the revised version is suitable for publication.

Reviewer #2 (Remarks to the Author):

The authors' due diligence in addressing the reviewers' comments is commendable and made for a better manuscript. This work should generate a lot of excitement in the field of 3D and multi-material printing.

Editorial's requests are in **bold**, the authors' responses are in Roman

Reviewer #1 (Remarks to the Author):

The authors have provided complete responses to all of the referee comments. I feel that the revised version is suitable for publication.

Response:

We are happy that we have addressed all comments of the referees.

Reviewer #2 (Remarks to the Author):

The authors' due diligence in addressing the reviewers' comments is commendable and made for a better manuscript. This work should generate a lot of excitement in the field of 3D and multi-material printing

Response:

We are happy that we have addressed all comments of the referees. We are indeed also very excited in publishing this work and in making an impactful contribution to the field of 3D and multimaterial printing.